# Robust Pixel Design Methodologies for a Vertical Avalanche Photodiode (VAPD)-Based CMOS Image Sensor

**DOI:** 10.3390/s24165414

**Published:** 2024-08-21

**Authors:** Akito Inoue, Naoki Torazawa, Shota Yamada, Yuki Sugiura, Motonori Ishii, Yusuke Sakata, Taiki Kunikyo, Masaki Tamaru, Shigetaka Kasuga, Yusuke Yuasa, Hiromu Kitajima, Hiroshi Koshida, Tatsuya Kabe, Manabu Usuda, Masato Takemoto, Yugo Nose, Toru Okino, Takashi Shirono, Kentaro Nakanishi, Yutaka Hirose, Shinzo Koyama, Mitsuyoshi Mori, Masayuki Sawada, Akihiro Odagawa, Tsuyoshi Tanaka

**Affiliations:** Panasonic Industry Co., Ltd., 1006, Oaza Kadoma, Kadoma-shi 571-8506, Osaka, Japan; torazawa.naoki@jp.panasonic.com (N.T.); yamada.shota@jp.panasonic.com (S.Y.); sugiura.yuuki@jp.panasonic.com (Y.S.); ishii.m@jp.panasonic.com (M.I.); sakata.yusuke@jp.panasonic.com (Y.S.); kunikyo.taiki@jp.panasonic.com (T.K.); tamaru.masaki@jp.panasonic.com (M.T.); kasuga.shigetaka@jp.panasonic.com (S.K.); yuasa.yusuke001@jp.panasonic.com (Y.Y.); kitajima.hiromu@jp.panasonic.com (H.K.); koshida.hiroshi001@jp.panasonic.com (H.K.); kabe.tatsuya@jp.panasonic.com (T.K.); usuda.manabu@jp.panasonic.com (M.U.); takemoto.masato3@jp.panasonic.com (M.T.); nose.yugo@jp.panasonic.com (Y.N.); okino.toru@jp.panasonic.com (T.O.); shirono.takashi@jp.panasonic.com (T.S.); nakanishi.kentaro@jp.panasonic.com (K.N.); hirose.yutaka@jp.panasonic.com (Y.H.); koyama.shinzo@jp.panasonic.com (S.K.); mori.mitsuyoshi@jp.panasonic.com (M.M.); sawada.masayuki@jp.panasonic.com (M.S.); odagawa.a@jp.panasonic.com (A.O.); tanaka.tsuyoshi@jp.panasonic.com (T.T.)

**Keywords:** single-photon avalanche diodes (SPADs), CMOS image sensors (CISs), guard ring, time-of-flight sensors, robust pixel design

## Abstract

We present robust pixel design methodologies for a vertical avalanche photodiode-based CMOS image sensor, taking account of three critical practical factors: (i) “guard-ring-free” pixel isolation layout, (ii) device characteristics “insensitive” to applied voltage and temperature, and (iii) stable operation subject to intense light exposure. The “guard-ring-free” pixel design is established by resolving the tradeoff relationship between electric field concentration and pixel isolation. The effectiveness of the optimization strategy is validated both by simulation and experiment. To realize insensitivity to voltage and temperature variations, a global feedback resistor is shown to effectively suppress variations in device characteristics such as photon detection efficiency and dark count rate. An in-pixel overflow transistor is also introduced to enhance the resistance to strong illumination. The robustness of the fabricated VAPD-CIS is verified by characterization of 122 different chips and through a high-temperature and intense-light-illumination operation test with 5 chips, conducted at 125 °C for 1000 h subject to 940 nm light exposure equivalent to 10 kLux.

## 1. Introduction

Single-photon avalanche diode (SPAD)-based CMOS image sensors (CISs) have reached a commercialization stage after demonstrations of mega-pixel devices [1,2,3], high-dynamic-range imagers [4,5], and a quanta imaging sensor [6]. Targeted applications of SPAD-CISs range from consumer products such as time-of-flight (TOF) ranging sensors for autonomous driving [7], face recognition cameras for mobile phones [8], and surveillance cameras [9], to advanced applications like fluorescence imaging in biology [10] and ultra-long-range systems for space developments [11,12,13]. Combined with cutting-edge CIS device technologies, such as back-side illumination, wafer bonding [14], and charge focusing structures [2,15], pixel scaling is realized without reducing sensitivity. Despite these advancements, the isolation structures between SPAD pixels, such as guard rings, limit the minimum pixel size to approximately 5 μm [16]. However, theoretical analysis on quenching circuit suggests that SPADs as small as 1 μm are feasible [17]. Consequently, recent studies have reconsidered guard-ring structures, including, guard-ring-sharing SPADs [18] and guard-ring-free vertical avalanche photodiodes (VAPDs) [19], although the design tradeoffs have not been discussed.

As for the transition to mass production, a critical issue is the robustness of SPAD pixels against important risk factors such as process variations, variations in applied voltage and temperature, and extreme illumination like intense sunlight. Variations in bias voltage and temperature can change device characteristics such as photon detection efficiency (PDE) and dark count rate (DCR) [20]. Although temperature compensation circuits have been developed for linear-mode avalanche photodiodes [21,22], these characteristic changes are not adequately addressed in SPAD-CISs.

Given that SPAD-CIS operation involves device electric fields orders of magnitude higher than those of conventional CISs, the risk factors such as process variation and extreme sunlight should be addressed at the pixel design level to ensure device robustness and reliability. Although reliability tests of SPADs under temperature stresses are reported [23,24,25], operational robustness against intensive illumination conditions and simultaneously added high-temperature conditions is not reported so far.

In this paper, we report on robust pixel design methodologies in our one-megapixel VAPD-based CIS (VAPD-CIS), targeted for a long-distance TOF ranging system [3,7]. Three important practical factors are considered: (i) a “guard-ring-free” pixel isolation layout, (ii) device characteristics (especially PDE and DCR) ”insensitive” to variations in applied voltage and temperature, and (iii) stable operation under intense light exposure conditions. For (i), we present a “guard-ring-free” layout for isolating VAPD pixels only by impurity implantation. A robust layout is demonstrated that resolves the tradeoff between pixel isolation and electric field concentration within the range of process variations. For (ii), we introduce a global feedback resistor (GFBR) connected to the substrate backside (the anode of all pixels) to suppress variations in PDE and DCR regardless of external voltage and temperature. Stable operation of the long-distance TOF ranging system [3,7] is verified across a chip temperature range from −25 °C to 105 °C, and the operational principle of the GFBR is discussed. For (iii), stable operation even under 100 kLux sunlight exposure is achieved by incorporating overflow transistors (OFTs). The robustness and reliability of VAPD-CISs are confirmed through high-temperature/intensity-light-illumination operation tests conducted at 125 °C for 1000 h subject to light exposure equivalent to 10 kLux.

## 2. Robust Pixel Design of “Guard-Ring-Free” VAPD-CIS

In this section, we present the pixel design methodologies for a “guard-ring-free” VAPD-CIS. In the VAPD-CIS, pixels are isolated solely by impurity implantation, which allows for the reduction in pixel size while maintaining a high fill factor. We put forward a robust pixel layout that both relaxes the electric field concentration and maintains pixel isolation even with the worst process variation.

### 2.1. Device Architecture of a VAPD Pixel Compared to a Conventional SPAD Pixel

We start with the device architecture of the VAPD pixel. Figure 1 illustrates schematic cross-sectional diagrams of (a) a conventional SPAD and (b) a VAPD pixel. The surface of the VAPD, acting as the cathode of VAPD, is a moderately doped n-type region formed on a p-type substrate, which acts as the anode. A buried PN junction acting as a multiplication region (MR) is formed away from the substrate surface. This plain structure of the VAPD can remove the guard ring, and isolation between neighboring pixels is achieved only by controlling impurity implantation. To minimize the isolation width while mitigating horizontal electric field intensity between the VAPD cathode and the isolation region, the isolation region is depleted by tuning the impurity density. This minimization of the isolation width enhances the fill factor compared to the conventional SPADs. In VAPDs, it is crucial to minimize surface defects in the isolation region to reduce dark current, and therefore, no contacts or trenches are formed in the isolation region. Thus, the isolation region is electrically floating, which is the uniqueness in the pixel isolation of VAPD-CISs. Additionally, the MR is positioned away from the substrate surface to prevent dark current resulting from surface defects manifesting as dark counts in the Geiger-mode operation.

### 2.2. Specifications and Chip Designs for VAPD-CISs

Table 1 summarizes the specifications and experimental results for the fabricated VAPD-CISs. These specifications are applicable across all temperatures, with some specifically designated at a room temperature (RT) value where the chip temperature (*T*_chip_) is assumed to be 40 °C. The specifications are determined by assuming application to the sub-range-synthesis TOF (SRS-TOF) system described in our previous works [3,7]. The sensor chips are fabricated using a 65 nm 1P4M1MIM (metal insulator metal) process, with a pixel pitch of 6 μm. The array size is 1200 × 900 pixels, and the frame rate is 30 fps. The fill factor, determined by the areal ratio of the N cathode to the pixel area, is 30.6% when the circuit area is included and 57.3% when the circuit area is excluded. To meet the rated voltage of the SRS-TOF system, the |*V*_BD_| specification is below 32 V. To detect objects with 10% reflectivity at 250 m away in each frame, PDE must exceed 1%. Due to the short exposure time, the DCR specification is not stringent and should be below 54,000 counts per second (cps)/pix, ensuring no counting error due to DCR occurs during the exposure periods.

In Figure 2, a chip photograph overlaid with circuit block diagrams (Figure 2a) and a pixel circuit of VAPD-CIS (Figure 2b) are illustrated [3,7]. The pixel circuit comprises a VAPD, six transistors, and an MIM capacitor. The anode voltage of the VAPD is supplied from a power supply (*V*_sub_) through an external global-feedback resistor (GFBR) to the substrate.

Figure 2c presents a timing diagram of a single frame for a Geiger-mode operation of the pixel circuit shown in Figure 2b. At the start of the frame, when RST and CNT are HIGH, FD and MIM are reset to the drain voltage (*V*_D_ = 3 V). Next, a predetermined number of repetition units, consisting of reset, exposure, and counting periods, are executed. During the reset period, OFTs are set to HIGH, resetting the cathode of each VAPD to *V*_D_. After the reset period, OFTs and TRNs are set to LOW and HIGH, respectively, during the exposure period. When a photo-charge is generated by a single photon during this period, a Geiger-mode multiplication is capacitively quenched, and the generated charges are accumulated on the VAPD and subsequently transferred to FD. The length of the exposure period is determined by the average distance of the measured subrange [3,7]. The charges in the FD are then transferred to the MIM capacitor during the counting period. After completing the series of gating actions, the accumulated charges are read through the source follower (SF) while SEL is set to HIGH. It is important to note that during the exposure period, the channel potential of the OFT is controlled to be higher than that of the isolation potential, releasing the excessive charges reaching the saturation potential through the OFT rather than allowing them to flow into neighboring pixels, FDs, or any other elements of the pixel. This effectively prevents blooming. Additionally, the pixel circuit is tolerant to Geiger-mode current during the reset period by suppressing it with the GFBR. The GFBR value is set to 1 kΩ. As explained in Section 3.2, owing to the GFBR, the voltage of the anode shared by the substrate is clamped, making the excess voltage applied to the VAPDs insensitive to variations in *V*_sub_ and temperature.

### 2.3. Potential and Layout Designs for Electric Field Relaxation and Pixel Isolation

Since the isolation region is floating in the VAPD pixels, its potential is position-dependent and electric field concentrates at the edges of the pixels. Therefore, ensuring a potential barrier that prevents charge overflow to neighboring pixels and relaxing the electric field concentration at the pixel edges are in a tradeoff relation. To solve the tradeoff, it is crucial for robust pixel operation to design the impurity density and the pixel layout. Figure 3 illustrates schematic top-view diagrams and simulation results of VAPD-CIS pixels. We consider two types of layouts: a 2 pixel rows per 1 circuit row (2P1C) layout (Figure 3a), and a 1 pixel row per 1 circuit row (1P1C) layout (Figure 3b). For clarity, both figures include two rows of VAPD pixels and surrounding circuits and the repetition unit is shown by a vertical arrow. In both layouts, N wells and P wells are arranged in straight lines, and the P wells are surrounded by the N wells to isolate P wells from P substrate. In the 2P1C layout, two VAPD rows are directly adjacent to each other, with circuit rows above and below them on the outside. In the 1P1C layout, on the other hand, rows of VAPDs and rows of circuits are arranged alternately. The fill factor is larger for the 2P1C layout since the circuit area for two rows of VAPDs is consolidated into one circuit row. It should be noted that the P isolations contain neither contacts nor trenches to avoid unnecessary defects.

The difference between the two types of layouts lies in the electric field in the horizontal direction near the intersections of the isolation regions calculated by a T-CAD simulation shown in Figure 3c,d. In these figures, the boundaries of the PN junction are represented by brown lines, and the edges of the depletion region are indicated by solid white lines. As can be seen in Figure 3c, for the 2P1C layout, the width of the depletion layer in the horizontal direction narrows near the intersection of the isolation regions, causing an electric field concentration at the edge of the VAPD pixels, as indicated by red-colored regions. This electric field concentration can lead to undesirable edge breakdown near the substrate surface, before forming a sufficient electric field in the MR, resulting in a lower PDE and higher DCR. Conversely, in the case of the 1P1C layout, the area between the neighboring pixels is completely depleted and the electric field is relaxed, as indicated by blue- or green-colored regions (Figure 3d). This is due to the N wells, which uniformly fix the potential in the isolation regions and prevent electric field concentration.

In Figure 4a, the potential profiles along the depth direction at points D, E, F, and G (indicated in Figure 3a,b) are illustrated, highlighting the position dependence of the potential within the isolation region. The potential differences at points D, E, and F compared to point G (at the substrate surface, depth = 0 μm) create barriers for pixel isolation. The potential is highest at point F, with a small difference from point G; therefore, isolation between pixels is the weakest at point F and we call the potential at this point the “isolation potential”. On the other hand, the potential is lower at points D and E, resulting in the strongest electric field near these points, forming indicators of edge breakdown.

In Figure 4b, the isolation potential at point F and the electric field intensity near points D and E are plotted as functions of the impurity density in P isolation, highlighting the tradeoff. When the impurity density is low, the isolation potential is insufficient to avoid charge overflow, and when the impurity density is high, electric field concentration causes edge breakdown. The specifications of both the isolation potential (<0 V) and the electric field (<420 kV/cm) are shown by orange and purple dashed lines, respectively. Calculated by considering the maximum voltage amplitude by quenching and the breakdown threshold, respectively. The minimum impurity density is defined by exceeding the maximum specification of the isolation potential (green vertical line), while the maximum impurity density is defined by exceeding the maximum specification of the electric field (red vertical line for the 2P1C layout and blue vertical line for the 1P1C layout). The acceptable range of impurity density for the 2P1C layout is too narrow (indicated by the red arrow), particularly considering that potential can vary by ±1 V due to process variations equivalent to 10% in the impurity density, as explained in Section 3.1. Therefore, we adopt the 1P1C layout that is more robust to the impurity density variation. It should be noted that in the 1P1C layout, PDE fulfills the specification of our TOF system as explained in Section 3.

## 3. Verification of the Robustness of VAPD-CIS

### 3.1. Verification of Potential Design against Process Variation

To validate the effectiveness of the above potential design, we fabricated pixel test element groups (TEGs) for I-V characteristic measurements where DC biases are directly applied to the pixel cathodes and the anode (substrate backside). The measurement results of the isolation potential compared to simulation results are depicted in Figure 5a with respect to impurity density. In the measurement, the isolation potentials are defined as the voltage at which current begins to flow between two adjacent pixels when the cathode voltage of one pixel is fixed and the voltage of the other pixel is decreased. To prevent a breakdown current, the absolute reverse voltage is maintained at 23 V during the measurement, slightly below the absolute value of the breakdown voltage (|*V*_BD_| = 27.5 V). Both the simulation and experimental results exhibit the same trend: a decrease in potential as the impurity density increases. The difference between simulations and experiments is due to the difference in definitions. In the simulations, the potential is directly observed, while in the measurements, the voltage value at which current begins to flow is obtained. In practice, the experimental definition is appropriate for characterization of pixel isolation as it directly reflects the overflow current. The measured breakdown voltages are independent of the impurity density, as illustrated in Figure 5b. This suggests that the electric field concentration is sufficiently relaxed, as edge breakdown near the intersections would cause the breakdown voltage to vary with the impurity density in the isolation region.

To verify the robustness of the isolation potential against process variations, we deliberately change the isolation width, as plotted in Figure 5c,d. In these figures, the horizontal axes are normalized such that the maximum process variation in the isolation width is 1. Even in the worst-case scenario for isolation potential, where the isolation width is reduced, it still meets the specified potential of 0 V or less (Figure 5c). Furthermore, the breakdown voltage remains constant, independent of the isolation width, indicating the absence of the edge breakdown. It should be noted that the variation in impurity density is negligible compared to that of the isolation width because the dose density is precisely controlled by the ion current. Therefore, the experimental error is mainly due to the isolation width and variations in the impurity density are shown only in Figure 5a,b.

Figure 6a,b present the cumulative probability plots of isolation potential and |*V*_BD_|, respectively. Each line represents the experimental results of 71 chips from the same wafer, and the experiments were conducted on seven different wafers. The average standard deviations of isolation potential and |*V*_BD_| within the same wafer are *σ*_P, intra_ = 0.19 V and *σ*_BD, intra_ = 0.24 V, while the standard deviations of the means of them between wafers are *σ*_P, inter_ = 0.078 V and *σ*_BD, inter_ = 0.062 V. Thus, the variations in the isolation potential are very small, enabling the robust design of VAPD pixels. It should be emphasized that the isolation potential and |*V*_BD_| do not exceed their maximum specifications for any chips measured, although the specifications are outside the bounds of Figure 6a,b (less than 0 V for isolation potential and less than 30 V for |*V*_BD_|).

### 3.2. Characteristics of Robustness to Applied Voltage and Temperature

In the following, robust device characteristics due to the GFBR are demonstrated. Figure 7a,b illustrate the experimental results of PDE and DCR in three temperature conditions. The exposure period is set to 300 ns for both PDE and DCR experiments. The PDE evaluations are conducted subject to an exposure condition of 100 nW/cm^2^ for an image plane with a 940 nm wavelength LED. The chip temperature (*T*_chip_) is controlled by a Peltier device. *T*_chip_ is set to −25, 40, and 105 °C, corresponding to environmental temperatures of −40, 25, and 85 °C, respectively, considering the 15- to 20-degree heat generation by the sensor operations. The error bars in Figure 7b indicate the standard deviation assuming a Poisson distribution. By applying a voltage exceeding |*V*_BD_|, both PDE and DCR increase with |*V*_sub_| due to the rise of |*V*_ex_| and avalanche triggering probability, and they saturate at approximately 1 V higher than |*V*_BD_| (|*V*_ex_| = 1 V). Although PDE decreases at low temperatures and DCR increases at high temperatures, they still meet the specifications indicated by black dashed lines. The plateau of the DCR in lower |V_sub_| region at *T*_chip_ = −25 °C is due probably to enhancement of impact ionization at low temperatures and to the existence of a temperature-independent dark current such as tunneling. It should be noted that the reasons why DCR at *T*_chip_ = −25 °C oscillates and that at *T*_chip_ = 40 °C gradually increases with |*V*_sub_| are also under investigation, but they should be due to small number of dark counts in the experiment; the numbers of dark counts are less than 100 out of a total of 5.4 million trials at each point.

Figure 8a–c present the temperature dependences of |*V*_BD_|, PDE, and DCR in comparison to the calculation results, respectively. The experimental results of PDE and DCR are taken from the plateau region where |*V*_ex_| > 1 V. The assumptions and formulas used for the calculations are detailed in Section A.1. |*V*_BD_| decreases with *T*_chip_, consistent with the calculated results. In Figure 8a, calculated and measured |*V*_BD_|’s show a similar trend except for a systematic offset. A possible explanation for this discrepancy could be the difference in the device structures of the VAPD-CIS and those used in the previous study, leading to the difference in their impact ionization coefficients [26]. PDE also increases with *T*_chip_ and aligns well with the calculations. Notably, the temperature dependence of PDE in the calculations is solely attributed to the photoelectric conversion efficiency of silicon [27], indicating that the avalanche triggering probability (ATP) is independent of temperature. This is due to the clamping effect of the GFBR fixing |*V*_ex_| without being influenced by |*V*_sub_| or *T*_chip_. The mechanism for eliminating the changes in |*V*_ex_| and ATP is explained in Section 5. In Figure 8c, the temperature dependence of DCR shows two components, i.e., an exponentially increasing (a black dashed line) component and a low-level offset (a gray dashed line) component. The former is attributed to originating from the diffusion current (as described by (A7) in Appendix A). The latter is not identified yet. However, one possible temperature-independent source would be a tunneling-related leakage current, as mentioned in Figure 7b. The discrepancy in DCR above *T*_chip_ = 40 °C can be attributed to errors in the calculation assumption and parameters used in the calculation, such as the diffusion coefficient, diffusion length, and intrinsic carrier density.

Figure 9a,b depict the accumulated frequency plots for DCR and PDE, respectively, at RT (*T*_chip_ = 40 °C) for two different wafers and a total of 122 chips. To prevent data variation due to the small number of counts, the slope of the counts from 300 ns to 500 μs is used as the DCR. Both PDE and DCR exhibit nearly negligible variances regardless of the wafers, with mean values of PDE = 1.45% and DCR = 21 cps/pix. Since the influence of |*V*_BD_| variation on the variations in PDE and DCR is suppressed by the GFBR, the standard deviations of PDE and DCR are small: 0.02% for PDE and 2.25 cps/pix for DCR. It should be noted that the difference in DCR values from those of Figure 8c is due to the different measurement definitions mentioned above.

## 4. Robust Operation of VAPD-CIS in High-Temperature and Intense-Light-Exposure Conditions

### 4.1. Stable Operation during Direct Imaging of the Sun (DIS)

To confirm the functionality of the OFT explained in Section 3.1, the robustness of the VAPD-CIS is tested by direct imaging of the sun during the daytime for three continuous hours. The illuminance values are measured every 1 h, and here they were measured to be >98 kLux (~100 kLux) during the entire period of the experiment. Considering that the field of view (FOV) of the system was 30 (±15 degrees) degrees, giving a cosine value of 0.97, the measured illuminance value was considered to be reasonable. Using the standard solar spectra [28], assuming uniform illumination of these photons, and taking into account the FOV of 30 degrees, the F number of the imaging lens set to unity, and the transmission coefficient of the bandpass filter at the wavelength of 940 nm, the average photon arrival rate to each pixel was estimated to be 1.4 × 10^10^ photons/s/pix. In Figure 10, an illustrative time-lapsed image (a) and actual images taken at each time after starting the experiment (b) are shown. During the entire period of the experiment, stable Geiger-mode operations of the VAPD device as well as the TOF camera were confirmed. For the pixels directly imaging the sun, the photon arrival rate was estimated to be ~200 times higher than the assumed uniform (or diffused) illumination. (This estimation is derived as an area ratio of the sun’s image (~500 pixels) to the total pixel area (~1 M pixels).) With such a high photon arrival rate, the amount of photogenerated charge due to sunlight exceeds the avalanche charge generated from just one photon. Thus, during imaging of the sun, a steady large current flowing through the VAPD pixels must be drained to the voltage source through the OFT even when the OFT is “OFF”. Figure 11a,b show, respectively, a schematic potential diagram and a corresponding equivalent circuit diagram for a VAPD and its periphery during the exposure period. The average current is estimated to be ~300 nA/pixel, which can be drained when *V*_gs_ = 0.6 V, as shown in Figure 11c. Thus, as shown in Figure 11a, by setting the gate voltage of the OFT to 1.1 V, the potential of the VAPD becomes 0.5 V, which is higher than the isolation potential between VAPD pixels (0 V). The excessive charges generated by the sunlight do not overflow to the neighboring pixels. We point out that with the Geiger-mode operation, one photon is enough to saturate the full-well capacity of the VAPD. Any amount of photocurrent generated after the Geiger-mode pulse generation and its quenching would contribute to the OFT current. It should be noted that without the robust design of the VAPD pixel described in Section 2, such stable operation would not be possible.

The main characteristics before/after the DIS experiment of the two tested chips are summarized in Table 2. The meanings of the acronyms are as follows: quantum efficiency (Q.E.) or sensitivity; the absolute value of the breakdown voltage (|*V*_BD_|); dark count rate (DCR); (d) photon detection efficiency (PDE); voltage amplitude of multiplication (AM); and variance of AM (σ_AM). These parameters are important not only as fundamental performance indices but also as robustness indices. They are sensitive to process variations and to conditions of temperature and illumination. Q.E. is measured with the normal CIS-mode operation, while the rest of the parameters are with the Geiger-mode operation. No significant change is observed for these parameters, satisfying the test specifications listed in the fifth column to detect device wear-out. Thus, the devices and the cameras are operated in stable conditions without any failure during the direct imaging of the sun.

Although the pixels directly imaging the sun receive about a 200-times-higher intensity than that of ordinary daylight (100 kLux), the numbers of these pixels are limited because the image of the sun is small, and the sun moves in the image plane of the VAPDs on cameras with fixed positions and view angles. Therefore, the above results also take account of the changes due to limited numbers of stressed pixels. In order to characterize only the exposed pixels, we conducted another similar experiment based on pixel-level analyses for three samples. Again, the camera was directly imaging the sun with the Geiger mode. The illumination condition was also 100 kLux, measured by the photometer. To correctly identify the exposed pixels, a slight zooming condition was employed. Because of the zooming condition, the intensity of the image was estimated to be about 1/8 of the experiment of Figure 11. To compensate for this, the exposure period was set to be longer, i.e., 60 min. In order to identify the exposed pixels, the sun was tracked during the experiment; the position of the sun was adjusted to be the center of the scene during the experiment (Figure 12). That is, the camera angle was adjusted such that 200 × 200 pixels at near center of the full-pixel area were continuously exposed during the entire experiment, as indicated by orange squares in Figure 12. Histograms of the exposed pixels taken with the normal mode under dark and illuminated conditions are shown in Figure 13. The histograms before and after the exposure are identical (overlapping) for both illumination conditions. Also, DCR measurement before and after the exposure did not show any change. These results were consistent with all three samples tested. So, we judge that no significant damage occurred in these pixels.

### 4.2. Stable Operation with No Degradation during High-Temperature/Intensity-Light-Illumination Operation Tests (HTs/ILOTs)

To demonstrate the robustness of the entire VAPD pixels at a high temperature and subject to strong light illumination, high-temperature/intensity-light-illumination operation tests (HTs/ILOTs) were conducted with five samples. During the test period, the devices were operated with the TOF (Geiger) mode [3,7], and the ambient temperature was set at a constant value of 125 °C to accelerate deteriorations. The duration of the test was 1000 h based on the requirement for AEC-Q100 Grade 1 [29]. A high-intensity light source with a wavelength limited to 940 nm was used, and the light intensity was set at 827 μW/cm^2^, estimated to be equivalent to 10 kLux illumination of sunlight [29], with the bandpass filter (BPF) at 940 nm ± 25 nm. The operation mode of the devices was a time-of-flight (TOF) mode with a 45,000 pulses/s exposure (gating) and a 450 fps reading cycle. These experimental conditions are summarized in Table 3.

In Figure 14, the principal performance indices characterized in the previous section (Table 2) of the VAPD-CIS are plotted as functions of the test period (time). As before, Q.E. is measured with the normal CIS-mode operation, while parameters from (b) to (f) are with the Geiger-mode operation. The maximum and the minimum test specification values are plotted as red and blue dotted lines, respectively. All the parameters show insignificant variation after the test period, remaining within the specifications. It should be noted that all the samples operated without errors during the test period. The constancy of Q.E., |*V*_BD_|, and PDE indicates stable three-dimensional dopant profiles inside the device. Insignificant changes in DCR, PDE, AM, and σ_AM_ are due to the robust pixel design described in Section 2.

## 5. Discussion

### 5.1. Mechanism of Voltage Stabilizing by GFBR for Robust Operation

As described in Section 3, the GFBR stabilizes |*V*_ex_| at a constant value, facilitating robust VAPD operation where PDE and DCR are insensitive to |*V*_sub_| and temperature. The mechanism of the voltage clamping by the GFBR is elucidated by examining the temporal sequence of potentials during the reset period. Figure 15 illustrates the schematic diagram of the potential diagram during the reset period (top) and the corresponding circuit diagram (bottom) in a time series from (a) to (e). The arrows indicate the current flowing through the VAPD pixels, and no current flows through the pixels without arrows. Figure 15a shows the potential diagram before the reset period, where charges are accumulated on the cathodes of VAPDs. Subsequently, in Figure 15b, immediately after the reset period starts when OFT is turned ON, the accumulated charges are discharged, fixing the cathode potential at *V*_D_, and then a dark current initiates avalanche multiplication. As illustrated in Figure 15c, avalanche current flowing through the OFT causes a voltage drop, fixing the reverse bias of the VAPD at |*V*_BD_|.

Since the capacitance on the anode side is the sum of all pixels, the capacitance on the anode side (represented in blue) is larger than those on the cathode side (represented in green) by one pixel. The voltage on the anode side remains almost unchanged during this period. As the number of VAPD pixels through which avalanche current flows increases, a large current flows through the GFBR, leading to a rise in the anode voltage, as depicted in Figure 15d. At this stage, quenching is enabled by OFT due to the reduction in *V*_gs_. Eventually, the cathode voltage returns to *V*_D_, as shown in Figure 15e, and |*V*_ex_| of the VAPD is stabilized at a constant value independent of |*V*_sub_| and temperature.

This mechanism has two requirements. First, a sufficiently long reset time is necessary to ensure the completion of the entire process. Second, the GFBR must be large enough to prevent the anode voltage from returning to |*V*_sub_| during the reset period. We experimentally determined that a reset time > 500 ns and GFBR > 1 kΩ are sufficient for this purpose. It should be noted that these values are consistent with the theoretical formulas of quenching dynamics, i.e., the reset time is longer than the time constant of quenching ((13) in [30]) and the threshold of GFBR is larger than the quenching resistance ((34) in [30]).

### 5.2. Temperature-Independent ATP: The Stabilizing Effect of GFBR

As shown in Figure 8b, the temperature dependence of ATP is completely suppressed in VAPD, although there is temperature dependence of impact ionization rates. This can be explained by the fact that a constant |*V*_ex_| leads to a constant ATP. Figure 16 illustrates the calculated impact ionization rates based on [26]. As shown in Figure 16a,b, the impact ionization rates are temperature dependent, leading to changes in |*V*_BD_| following the breakdown condition ((16) in [20]). This temperature dependence of the impact ionization rate is eliminated by normalizing to |*V*_BD_|. Figure 16c,d show the impact ionization rates as a function of *V*_ex_, where all curves align on a single curve. In these graphs, the positive direction of *V*_ex_ is the direction in which the reverse bias increases. Notably, the scaling of impact ionization rates can be explained by the fact that the breakdown condition ((16) in [20]) is temperature independent and, therefore, the impact ionization rates at |*V*_BD_| remain the same at all temperatures. Figure 16e presents ATP with respect to *V*_ex_ at each temperature, calculated using formula (A5) in Section A.1. It is evident that ATP consistently maintains nearly the same value regardless of temperature. By integrating the findings from Section 5.1 and Section 5.2, the temperature-independent APT is explained, providing evidence for the stabilizing effect of the GFBR.

## 6. Conclusions

We have presented robust pixel design methodologies for VAPD-CIS pixels that address process variations, applied voltage, temperature, and intense light exposure. Firstly, we demonstrated the potential design for “guard-ring free” VAPD pixels that are robust to process variations. In the 1P1C layout, the linearly arranged wells uniformly stabilize the potential in the isolation region, thereby preventing electric field concentration near the substrate surface (Figure 3). By setting the optimal impurity density, we resolve the tradeoff between pixel isolation potential and electric field concentration, within the range of isolation width variations (Figure 4 and Figure 5). The validity of the potential design was confirmed for the fabricated TEGs with seven different wafers (Figure 6). Secondly, robustness to applied voltage and temperature is achieved by the GFBR, connected to the anodes of all pixels, which maintains a constant |*V*_ex_| and ATP regardless of external voltage (|*V*_sub_|) and chip temperature. Variations in PDE and DCR are suppressed (Figure 7 and Figure 8), ensuring that all tested chips meet the specifications for our SRS-TOF system (Figure 9). The mechanism and limitations of the voltage stabilization by the GFBR were discussed based on the potential and current during the reset period (Section 5.1). Thirdly, OFTs are incorporated within each pixel to discharge excess charges generated by intense light exposure. Stable TOF (Geiger-mode) operation, even under direct imaging of the sunlight, was confirmed (Table 2). Finally, we conducted HTs/ILOTs for 1000 h at 125 °C subject to 940 nm light exposure, equivalent to 10 kLux. No significant degradations were observed in the main characteristics before and after the test (Figure 14), demonstrating the robustness and reliability of VAPD pixels. These advanced design methods can be integrated with other SPAD technologies to expand the range of applications for SPAD-CISs.

## Figures and Tables

**Figure 1 sensors-24-05414-f001:**
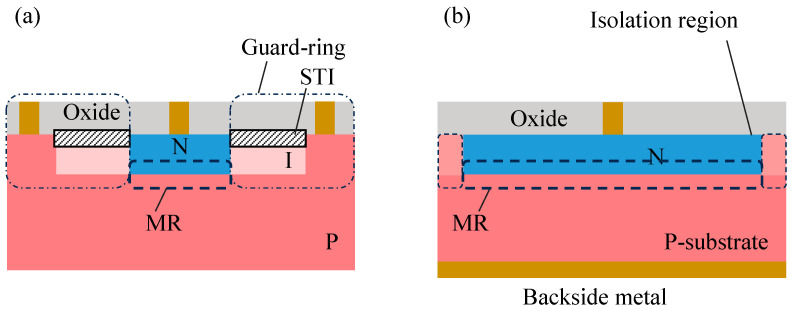
Cross-sectional views of a pixel: (**a**) a conventional SPAD and (**b**) a VAPD-CIS. N-type and P-type regions are drawn by blue and red, respectively.

**Figure 2 sensors-24-05414-f002:**
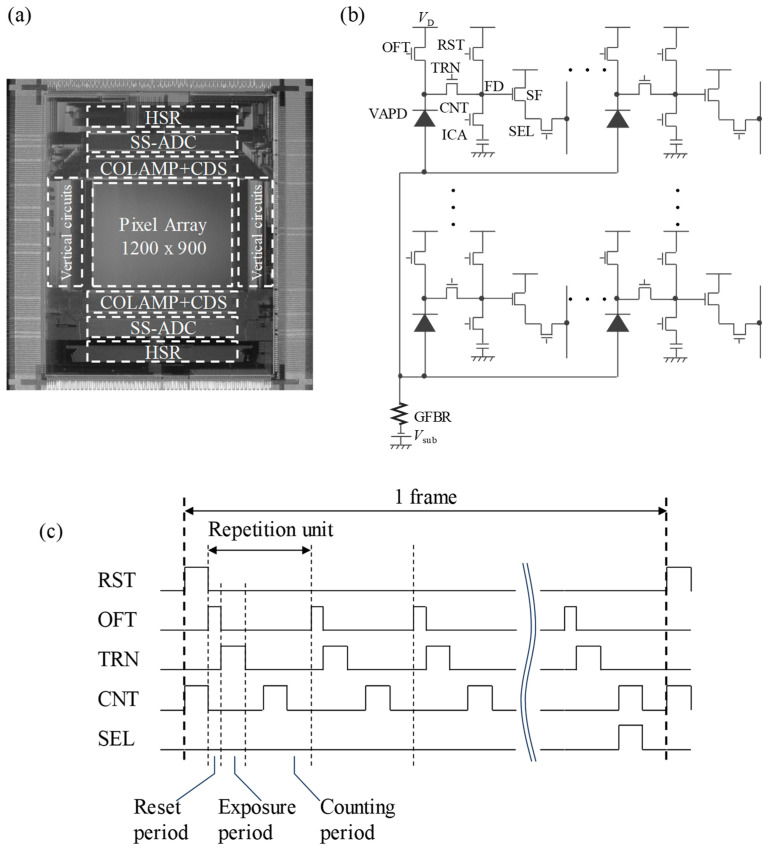
(**a**) A chip photograph of VAPD-CIS overlaid with circuit block diagrams. (**b**) A circuit diagram of the VAPD pixel array. (**c**) A schematic timing diagram of the pixel circuit illustrated in (**b**).

**Figure 3 sensors-24-05414-f003:**
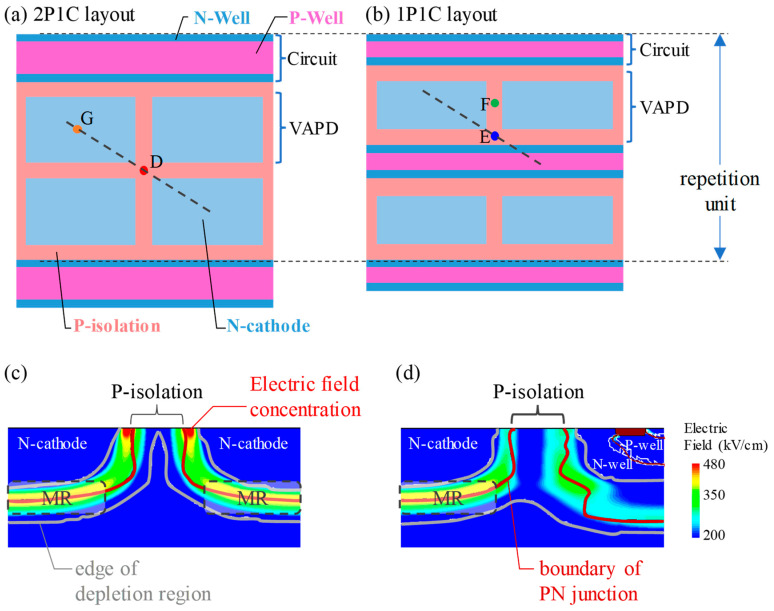
Top-view layout of the VAPD-CIS pixels: (**a**) 2P1C layout and (**b**) 1P1C layout. N-cathodes, N-wells, P-isolations, and P-wells are drawn by light blue, blue, magenta, and light orange, respectively. (**c**,**d**) Cross-sections of electric field intensity along the black dashed lines in (**a**,**b**). The electric field concentration in (**c**) is indicated by red-colored regions.

**Figure 4 sensors-24-05414-f004:**
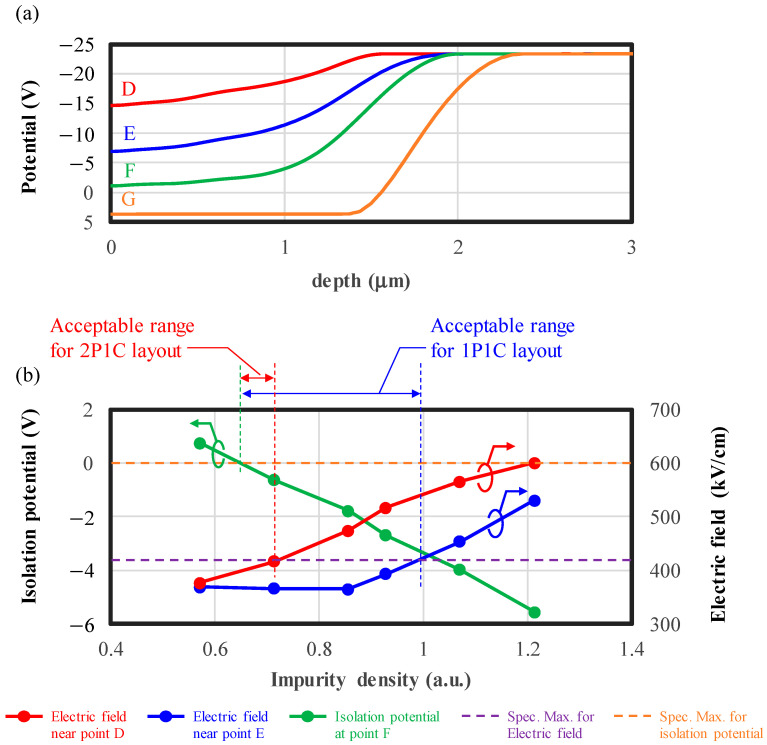
(**a**) Electrostatic potentials in the depth direction at points D, E, F, and G specified in Figure 3. (**b**) Electric field intensities near points D and E and the isolation potential at point F. The horizontal axis is normalized so that the typical impurity density is 1. The red and blue horizontal arrows indicate the acceptable ranges of impurity density for the 2P1C layout and the 1P1C layout, respectively.

**Figure 5 sensors-24-05414-f005:**
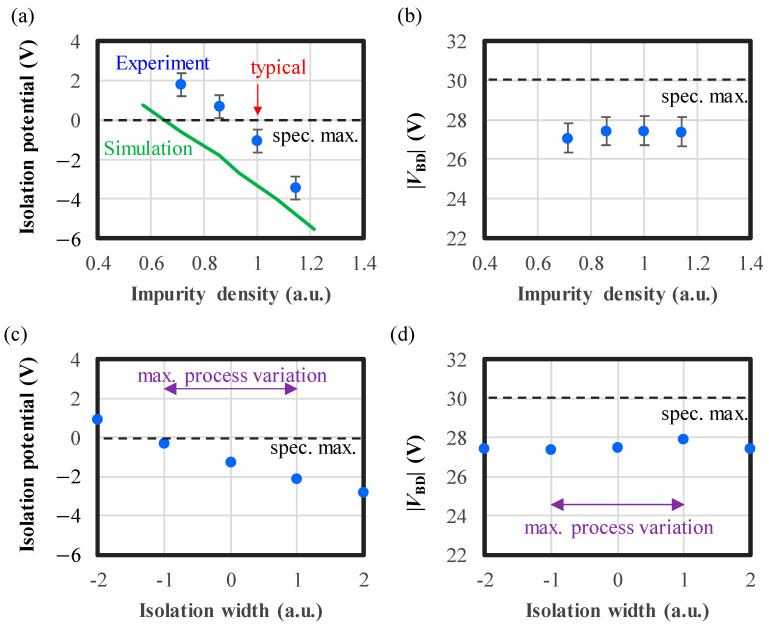
(**a**) Isolation potential and (**b**) breakdown voltage (|*V*_BD_|) with respect to the impurity density in the isolation region. The error bars indicate experimental ± 3*σ*. The horizontal axes are normalized so that the typical impurity density is 1. (**c**) Isolation potential and (**d**) |*V*_BD_| measured as a function of isolation width. The horizontal axes are normalized with the maximum variation in isolation width, as depicted by the purple arrows.

**Figure 6 sensors-24-05414-f006:**
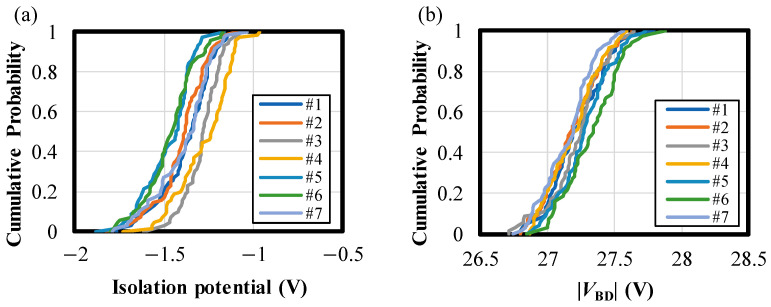
Cumulative probability plots as functions of (**a**) isolation potentials and (**b**) |*V*_BD_|. The plots with different colors correspond to the results from 7 wafers.

**Figure 7 sensors-24-05414-f007:**
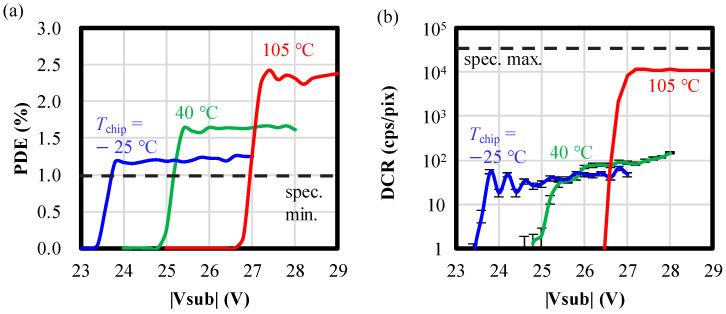
|*V*_sub_| dependences of (**a**) PDE and (**b**) DCR. The blue, green, and red lines denote the results for *T*_chip_ = −25, 40, and 105 °C, respectively. The black dashed lines indicate the maximum specifications.

**Figure 8 sensors-24-05414-f008:**
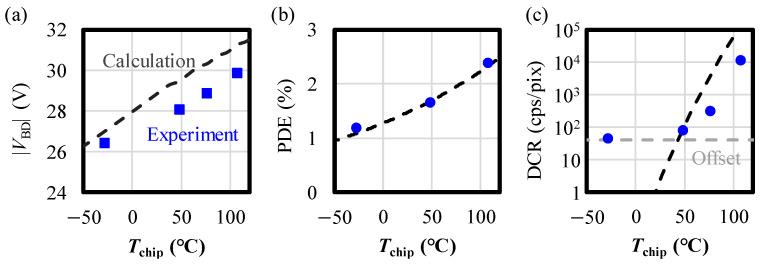
Temperature dependences of (**a**) |*V*_BD_|, (**b**) PDE, and (**c**) DCR. The black dashed curves and the blue dots represent the calculation and experimental results, respectively. The error of experimental points is within the size of the data markers (<10%).

**Figure 9 sensors-24-05414-f009:**
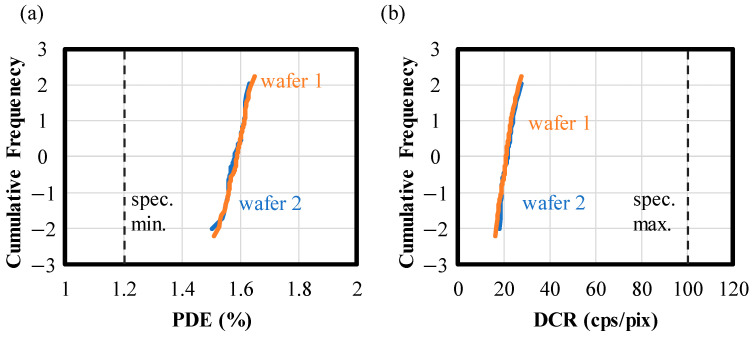
Cumulative frequency plots for (**a**) PDE and (**b**) DCR. The vertical axes are set to 0 when PDE or DCR takes the average value and to 1 when it deviates from the mean by standard deviations. The orange and blue lines denote the result for wafer 1 and wafer 2, respectively. The black dashed lines represent their specifications for RT.

**Figure 10 sensors-24-05414-f010:**
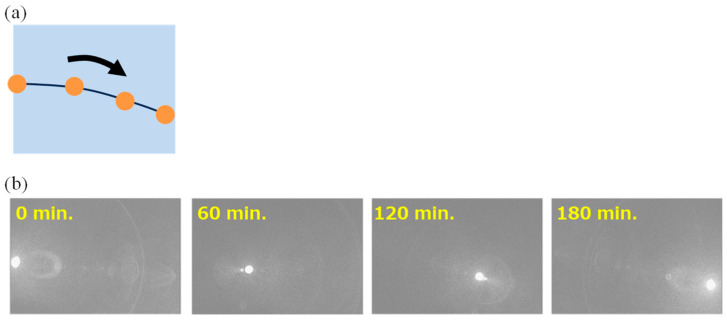
(**a**) An illustrative time-lapsed image of the sun. (**b**) Actual images of the sun taken at each time after starting the experiment. The test lasted for three hours, and as time passed, the sun, initially visible on the left edge of the screen, moved to the right.

**Figure 11 sensors-24-05414-f011:**
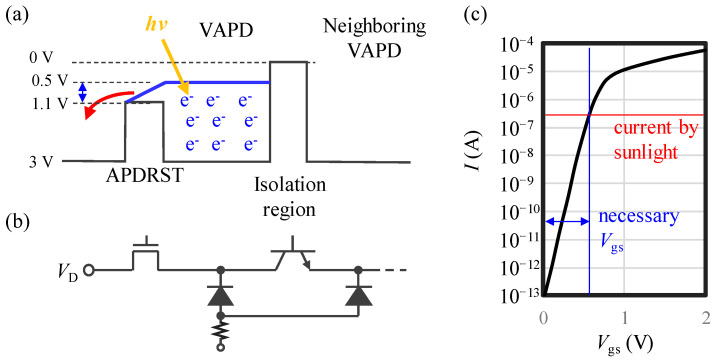
Schematic diagrams of (**a**) potential and (**b**) corresponding circuits during the exposure period. An NPN bipolar transistor indicates the isolation region. The yellow arrow represents incoming photon flux. (**c**) An I-V characteristic of the OFT. The red line denotes the assumed current by directly imaging the sun and the blue arrow represents necessary *V*_gs_.

**Figure 12 sensors-24-05414-f012:**
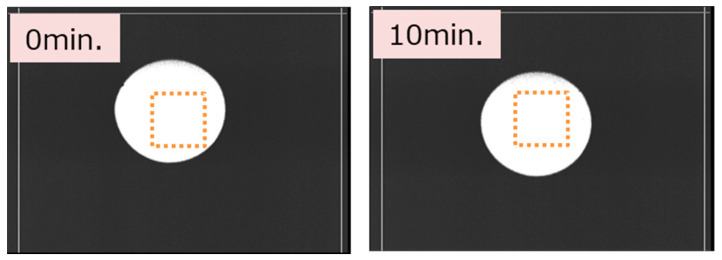
Images of the sun taken at the initial stage (**left**) and 10 min after the start of the experiment (**right**). The area enclosed by orange dotted squares represents the pixels continuously imaging the sun during the exposure period (60 min).

**Figure 13 sensors-24-05414-f013:**
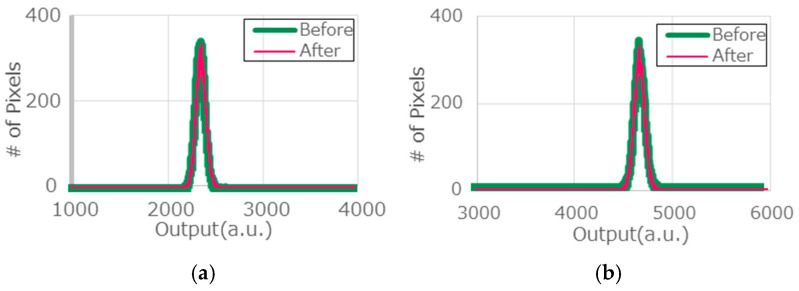
Histograms of output signals from the pixels continuously imaged the sun extracted in Figure 12 before and after the direct sun-imaging experiment. (**a**) Dark condition and (**b**) bright illumination condition.

**Figure 14 sensors-24-05414-f014:**
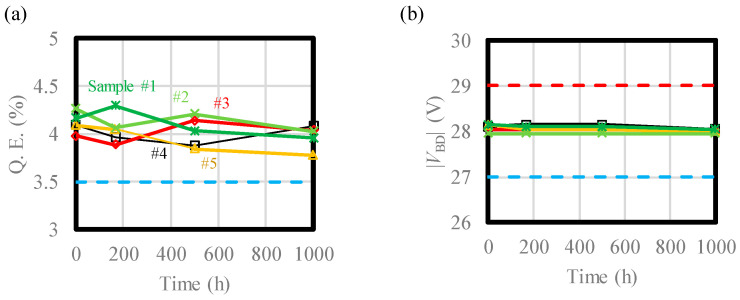
Performance indices of the devices tested by direct imaging of the intense light source as functions of time: (**a**) Q.E., (**b**) |*V*_BD_|, (**c**) DCR, (**d**) PDE, (**e**) AM, (**f**) σ_AM_.

**Figure 15 sensors-24-05414-f015:**
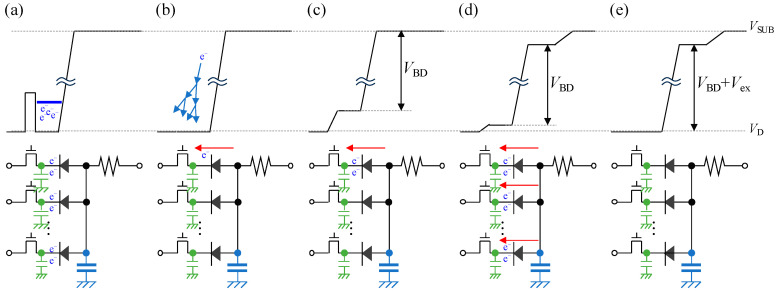
Schematic potential diagrams (**top**) and their corresponding circuits (**bottom**). Subfigures (**a**–**e**) illustrate a time series of states: (**a**) prior to the reset period, (**b**) at the start of the reset period, (**c**) a period when the current flows through a single VAPD, (**d**) after a large current is generated, and (**e**) after the cathode of the VAPDs returns to *V*_D_.

**Figure 16 sensors-24-05414-f016:**
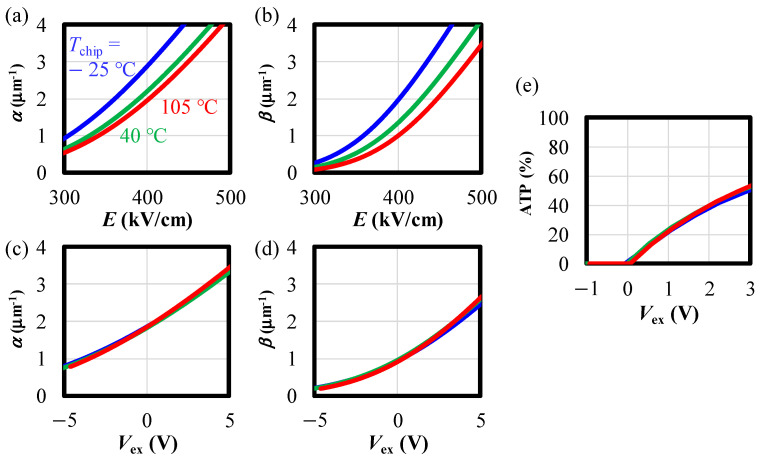
Impact ionization rates for (**a**) electrons and (**b**) holes, as a function of the electric field. (**c**,**d**) depict the same data with *V*_ex_ as the horizontal axes. (**e**) ATP as a function of *V*_ex_. The red, green, and blue curves represent the results at *T*_chip_ = −25, 40, and 105 degrees, respectively.

**Table 1 sensors-24-05414-t001:** Specification sheet of fabricated VAPD-CISs.

Items	Specifications	Typical Experimental Results
CMOS technology	65 nm 1P4M1MIM	
Pixel pitch	6 μm	
Array size	1200 × 900	
Frame rate	30 fps	
Fill factor	30.6% (including circuit area) 57.3% (excluding circuit area)	
|*V*_BD_|	<32 V	29.8 V @−25 °C 27.5 V @RT
Isolation potential	<0 V	−1.3 V @RT
PDE @940 nm	>1% >1.2%@RT	1.2% @−25 °C 1.6% @RT
DCR	<100 cps/pix @RT <54,000 cps/pix @105 °C	21 cps/pix @RT 10,000 cps/pix @105 °C
Voltage swings	>1 V	1.5 V

**Table 2 sensors-24-05414-t002:** Summary of main characteristics before/after DIS experiments (*n* = 2).

Item	Before	After	Variation	Test Spec.
Q.E. (%)	4.38%	4.20%	−4.0%	3.5% max.
|*V*_BD_| (V)	28.15 V	28.15 V	0%	27 V~29 V
DCR (cps)	21.5 cps	20.4 cps	−5.0%	50 cps max.
PDE (%)	1.46%	1.46%	0%	1% min.
AM (V)	1.50 V	1.50 V	0%	1 V min.
*σ*_AM_ (V)	0.083 V	0.084 V	1.4%	0.17 V max.

**Table 3 sensors-24-05414-t003:** Test conditions for HTs/ILOTs.

Temperature	125 °C
Illumination power	827 μW/cm^2^
Wavelength	940 nm
Exposure Time	1000 h.
Optical Filters Condition	w. BPF and Lens
Driving mode	Time-of-Flight (45,000 exposures/s)
Read-out speed	450 fps

## Data Availability

Data are unavailable due to privacy restriction.

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
