# Peer review of "Robust Pixel Design Methodologies for a Vertical Avalanche Photodiode (VAPD)-Based CMOS Image Sensor"

_sensors, 2024, doi:10.3390/s24165414_

Round 1

Reviewer 1 Report

Comments and Suggestions for Authors

This paper proposes two main points: firstly, a guard-ring-free layout method to enhance fill factor, and secondly, the use of a global feedback resistor to improve CMOS image sensor performance under intense light. The test results are extensive, highlighting stable performance and good reliability. There is a lack of theoretical analysis of the innovation points and research background investigation. Here are the specific questions

1、The abstract is written strangely; this section currently outlines your work and should be placed as the last paragraph of the introduction.The abstract should be shorter and should not be written in points.

2、Usually, abbreviations and explanations are not used in the abstract; full terms should be used.

3、There are many abbreviations used in the paper, some of which are unnecessary, such as BSI, W/B, etc. Because they are not mentioned again. Please check.

4、The introduction lacks a discussion of other researchers' findings on this topic. The paper review work is missing.

5、In the last paragraph of the introduction, "IR" is mentioned without an explanation of its meaning.

6、There is inconsistent font formatting in the title of the second section.

7、"VPD" in line 86 appears to be miswritten.

8、"tradeoff" and "trade-off" are used in the paper. Should they be consistent?

9、The definitions of Figure 2(a) and 2(b) are not well distinguished. I believe the distinction lies not in a symmetrical approach, but rather in the layout of the P-Well.

10、The green curve in Figure 3(b) is not labeled.

11、The "TEGs" in line 180 are not explained in full in the paper.

12、Is the title of the figure too long? It is repetitive with the descriptions in the paper, not concise, and a struggle to read.

13、The coordinates in Figure 5 are set up strangely, and it's hard to see the difference between the curves.

14、Is “Sensor” in line 230 duplicated with the “sensor” in CIS? There is the same query below.

15、The terms ‘RT’ and ‘cps/pix’ in Table 1 are not explained in the paper.

16、Figure 6 is not labeled with the figure number of Figure 6(b), and there are no FD nodes.

17、The typeface in Figure 7 should not be larger than the typeface of the figure title. The width of curves should not coincide with the width of the axes.

18、The reasons for the large differences between calculated and experimental values in Figures 8(a) and 8(c) are not mentioned.

19、Is Figure 9 missing the determination of temperature?

20、The meaning of ‘FOV’ in line 335 is not explained.

21、What does the ‘F’ in line 336 mean?

22、Is the number of samples missing from the description in Figure 12? Fig. 12(a), Fig. 12(d), Fig. 12(e) The axes can be set more reasonably so that each curve is visible.

23、Is there an error in line 465 where the formula points to something that can't be found?

24、References 25-30 are old and, [25] and [26] are not cited in the paper.

25、Inconsistent reference formatting needs to be checked.

Comments on the Quality of English Language

I recommend to check the english

Reviewer 2 Report

Comments and Suggestions for Authors

Full review is found in the attached word document

Comments on the Quality of English Language

Overall the quality of the English is good, only minor tweaks to some parts of the text are required.

Round 2

Reviewer 2 Report

Comments and Suggestions for Authors

All comments are found in the word document

Author Response

Thank you very much for taking the time to review our manuscript. We have addressed your comment and have made the necessary revision. 

Comment: Line 361 – I think this should read “This estimation is derived as an area ratio of the sun’s image…” 

Response: Thank you very much for your comment. We have corrected "areal" to "area" in line 361 as you pointed out.